# OpenReview forum: "Adversarial Arena: Crowdsourcing Data Generation through Interactive Competition"
_ICLR.cc/2026/Conference — Submitted to ICLR 2026_

### Official Review · Reviewer_q1j2 · 2025-10-26

**Soundness:** 2
**Presentation:** 2
**Contribution:** 1
**Rating:** 2
**Confidence:** 4

**Summary:**

This paper introduces **Adversarial Arena**, a public platform for generating high-quality, diverse synthetic data through structured, multi-turn adversarial competitions between independent teams of attackers and defenders. The core idea is to frame data generation as a competitive task: attackers attempt to elicit failures (e.g., generating unsafe code), while defenders aim to produce robust, correct responses. This interactive process, orchestrated over multiple tournament rounds, naturally produces complex, multi-turn conversational data.

**Strengths:**

* The paper proposes **Adversarial Arena**, a public competition platform capable of generating high-quality data, and clearly elaborates on various details regarding the platform's construction, operation, and data generation process.
* Based on this public competition platform, the paper constructs a dataset for cybersecurity alignment, and experiments demonstrate the promising fine-tuning performance achieved using this dataset.

**Weaknesses:**

* The academic contribution of this paper is limited. The authors primarily emphasize the promising fine-tuning performance of their constructed dataset for safety alignment. However, they neither propose a new challenge (e.g., novel problems or paradigms within the safety alignment domain) nor deliberately construct data to address a specific, existing problem. While diversity is emphasized, the authors fail to provide a detailed explanation of what specific diversities were achieved. Furthermore, the authors act primarily as platform builders; the contributions of the participants and the details of their solutions lack systematic elaboration and summarization.
*   The experiments in this paper are insufficient. For instance, the fine-tuning experiment in Table 2 lacks baseline comparisons. It would be crucial to compare against recent or classic datasets for cybersecurity alignment to understand how fine-tuning on those datasets impacts a model's resilience to attacks.
*   The analysis of data diversity is somewhat superficial. Relying solely on t-SNE visualizations or embedding similarities to measure diversity and bias cannot effectively reveal novel problems or paradigms. (Even the same security issue, when phrased differently, can lead to significant shifts in vector representations). The authors should focus more on summarizing what new security issues or novel attack paradigms emerged from the competitive interactions among participants on their platform.

**Questions:**

Consistent with the Weaknesses section, I do not recommend the authors to proceed with a rebuttal. If they choose to do so, please just provide targeted responses addressing the points raised in the Weaknesses.

---

> ### Author Response · Authors · 2025-11-20
> **Rebuttal response**
>
> Dear Reviewer, we are grateful for your time and effort put in reviewing our paper and greatly appreciate your comments. Thank you for noting that our method is capable of generating high-quality data and the fact that our paper presents details about the platform, construction, operation, and data generation process. We are also glad that you see the fine-tuning performance of our dataset to be promising.
>
> Below are our responses to the three weaknesses you pointed out:
>
>
> 1. As a reminder, the primary contribution of this work is NOT the fine-tuning performance of the datasets collected in the cybersecurity case study. Rather, the goal was to showcase a general framework for collecting diverse data, which removes the need to hand-engineer a task-specific diversity metric that can account for high-level semantic factors like attack strategy. Another core contribution of the paper is a new multi-turn Cybersecurity Alignment dataset, constructed using the adversarial arena and to be released publicly upon acceptance.
> 2. We do not include a baseline comparison in our experiments because, to the best of our knowledge, no relevant training datasets were publicly available at the time of submission. This is precisely where our framework is most useful: it enables data generation in domains where data does not yet exist or is too limited in scope.
> 3. Our objective was to demonstrate that the iterative, two-sided competition drives participating teams to explore different regions of the dialog space independently. The t-SNE plots were intended to illustrate this point. Additionally, the teams were tasked with inducing vulnerable or malicious code, rather than discovering new security issues, so we do not claim novelty in attack paradigms. Our diversity claims are about the semantic and syntactic variation in attacks generated under the Adversarial Arena framework.
>
> We hope these clarifications address your concerns and would appreciate it if you could revisit your score for this submission.

---

### Official Review · Reviewer_gzkS · 2025-10-27

**Soundness:** 4
**Presentation:** 3
**Contribution:** 2
**Rating:** 4
**Confidence:** 4

**Summary:**

This paper introduces Adversarial Arena, a framework for crowdsourcing high-quality and diverse LLM training data through adversarial competitions between teams. The key contributions are:

1. Adversarial Arena, a general framework where "defenders" aim to make models perform well on tasks of interest while "attackers" try to elicit failures, with multiple tournaments allowing iterative improvement
2. An instantiation of this framework for cybersecurity tasks with 10 university teams (5 attackers, 5 defenders) competing over 4 rounds
3. A resulting dataset of ~19k labeled multi-turn conversations that, when used for fine-tuning, improves secure code generation by 18% and cyberattack assistance refusal by 24% (on CyberSecEval benchmarks)

**Strengths:**

1. **Interesting approach to data collection**: The adversarial competition framework presents an intriguing method for obtaining training data. While similar competition-approaches have been used before, the systematic application to data collection beyond adversarial tasks is novel.
2. **Good case-study design**: Given the complexity of organizing and orchestrating a multi-team competition, the case-study itself is a strong contribution. In particular, the authors also provide an orchestrator for such competitions with good documentation.
3. **Successful empirical validation**: The case study demonstrates practical effectiveness, with the resulting dataset yielding meaningful improvements on the cybersecurity task of interests. The validation methodology through fine-tuning experiments is sound, and the approach to testing diversity via semantic alignment is reasonable.
4. **Good presentation**: The paper is well-organized with helpful visualizations (Figure 3) and generally clear. The framework's instantiation for cybersecurity (Section 4), except for minor parts of the setup, is documented in depth.

**Weaknesses:**

**Main points**

1. **Lack of truly adaptive attacks**: The most significant limitation is that attackers and defenders interact simultaneously rather than sequentially. The paper mentions this limitation on L428; however, it only mentions either fully online settings or more frequent tournaments as a solution. In practice, defenders must commit first, allowing attackers to adapt. Hence, a more realistic approach would be turn-based, such that attackers get access to all defenses before submitting an attack strategy. The current framework hence artifically limits attackers (and thereby data quality), and it is not clear to me if the orchestrator can easily be adapted to such a turn-based setting.
2. **Scalability concerns due to manual labor**: The approach requires substantial effort from many participants (e.g., 10 academic teams in the case-study). It's unclear whether this scales to the diverse set of tasks current LLMs have to support, thus limiting the practical impact of Adversarial Arena. For example, one big competition might see a lot of participation from a diverse set of teams, but multiple weekly competitions that span many tasks might see quickly diminishing interest.

**Minor issues**

3. **High-level framework with limited novelty**: The framework itself (Section 3) is relatively abstract; it could be helpful to make it more prescriptive. The paper's significant contribution, in my opinion, is more the instantiation of this framework on cybersecurity tasks and the resulting dataset/code, not necessarily the high-level approach. In particular, using an attacker-defender setting to obtain data or improvements has been explored before (e.g., [Bartolo et al., 2020](https://arxiv.org/abs/2002.00293), [Debenedetti et al., 2024](https://arxiv.org/abs/2406.07954), or the [Generative AI Red-Teaming Challenge](https://humane-intelligence.org/get-involved/events/defcon-2023-overview/)). The core idea of instantiating a general adversarial framework including *both* attackers and defenders is still novel and has, to the best of my knowledge, not been done before. Nevertheless, I think contextualization with existing approaches would be appropriate.
4. **Diversity measurement limitations**: The diversity penalty for the case-study (L298-L307) captures only lexical diversity through BLEU scores, not semantic diversity of attack strategies. While Section 4.3 analyzes semantic diversity post-hoc, this isn't incorporated into tournament scoring; doing so would potentially better align incentives.
5. **Mixed task evaluation**: The case study combines two distinct tasks (eliciting vulnerable code and cyberattack assistance), leading to attackers focusing on the easier target (see e.g., the last paragraph of Section 4.4). Separate tournaments or independent grading per task could avoid such exploitation.
6. **Limited orthogonality testing**: Lines 182-184 emphasize testing utility on orthogonal tasks, but evaluations only examine related coding/cybersecurity tasks rather than truly orthogonal capabilities like general knowledge or reasoning. Even if the model should only work for coding tasks, I believe defenses should still be evaluated on general-purpose coding questions truly orthogonal to cybersecurity.
7. **Missing details of case-study setup**: While the evaluations of the case-study are described in detail, some points about the setup were unclear to me; see questions.

**Questions:**

1. Do the authors plan to publicly release the collected dataset and orchestration code? This would significantly enhance the contribution's impact.
2. Can the framework be modified to support turn-based interactions where defenders release their defenses first, allowing attackers to adapt? This would better reflect real-world scenarios and potentially yield stronger attacks (and hence better data).
3. Were defender teams allowed to fine-tune the ChallengeLLM, or were they restricted to prompting and auxiliary models?
4. Is the 45-second latency budget (L244) per conversation, per completion, or aggregated differently? Does this constraint apply to both attackers and defenders?
5. Does each tournament consist of single or multiple conversations per attacker-defender matchup?

---

> ### Author Response · Authors · 2025-11-20
> **Rebuttal response (1/2)**
>
> Dear Reviewer, thank you for your thorough and insightful review. We are glad you found our data collection approach interesting and novel, use case design, and evaluation and liked the presentation of our paper. We are also happy that you found our case study and the empirical validation of the data collected from it compelling.
>
> We hope that our responses below can help elaborate on the contributions of our work. Our responses to your Questions are as follows:
>
> 1. Yes, we will release the dataset collected from the challenge on acceptance.
> 2. Our framework does support sequential turn based interactions.
>     * All conversations are multi-turn where each team receives the previous turn from the opponent before generating their response. So teams have the opportunity to adapt to their opponent’s behavior.
>     * Adversarial Arena also allows teams to adapt their strategy over conversations. This is because the framework only parallelizes conversations against different opponents.
>     * In the case-study we also introduced a probing phase with 200 conversations for attackers to understand the behavior of defenders, followed by another 200 scored conversations. This in essence causes defenders to reveal their defenses first and attacking systems can adapt to them.
> 3. Defender teams were allowed access to the full weights of the model and they were allowed to do continued pre-training and post-training on the models. We also allowed the use of small input and output guardrails with the model. Appendix C has some description of the novel approaches participating teams ended up trying.
> 4. The 45 second latency budget was per turn, where each conversation could have up to 10 turns (5 from the attacker and 5 from the defender). And yes, the latency budget for attackers and defenders was the same.
> 5. Each tournament in the case study consisted of 2 phases per attacker-defender match-up, each with 200 conversations. Conversations in the first phase were unscored were unscored where the attacker could probe the defender for weaknesses and adapt their attacks accordingly. Conversations in the second phase were counted towards scoring.
>
> We would also like to take this opportunity to clarify some of your other concerns:
>
> 1. **Lack of truly adaptive attacks:** The Adversarial Arena framework has both sequential and parallel interactions. Every turn in each conversation is generated sequentially. Also, each tournament has multiple conversations between each attacker-defender matchup sequentially. Attackers had three different opportunities to adapt
>     * *Automated adaptation of their system based on the probing phase:* Attackers were allowed to host adaptive systems that modify their approach based on previous interactions, while defenders could only host their fine-tuned version of the ChallengeLLM that could not adapt to attacks automatically.
>     * *Dynamic adaptation during interaction:* Attackers were also allowed to adapt to defenders’ behavior through the 200 conversations in each scored phase.
>     * *Adaptation of approach from tournament to tournament:* Attackers also had the opportunity to build new attack approaches between tournaments.
> 2. **High-level framework with limited novelty:** Thank you for sharing this relevant work, we will contextualize section 3 of our paper with these existing approaches in the final draft. As you pointed out that our method is novel due to both attackers and defenders being part of the adversarial framework. We would also like to point out some additional novel properties of our framework
>     * *Iterative framework:* Our framework provides opportunities for both attackers and defenders to improve their attacks/defenses leading to a shift in the distribution of data over tournaments. This leads to an overall richer dataset. Hence, while [Bartolo et al., 2020](https://arxiv.org/abs/2002.00293) discuss the use of their framework to generate training data, their framework doesn’t inherently lead to rich datasets like ours. Similarly, the frameworks proposed in the [Generative AI Red-Teaming Challenge](https://humane-intelligence.org/get-involved/events/defcon-2023-overview/) and the one introduced by [Debenedetti et al.,](https://arxiv.org/abs/2406.07954) do not give defenders the opportunity to improve over the challenge.
>     * *Scalability with automated attackers:* All existing adversarial frameworks have humans in the loop as attackers which makes them much harder to scale up.

---

> ### Author Response · Authors · 2025-11-20
> **Rebuttal response (2/2)**
>
> 3. **Diversity measurement limitations:** While we agree that BLEU based metrics only capture lexical diversity, the diversity incentive was specifically designed to avoid a narrow failure mode where attacker find an attack that works and keep repeating it to increase their score **in a single tournament**. For the overall dataset, the Adversarial Arena framework lets us overcome the difficult problem of designing a good diversity metric that can distinguish between different attack strategies because of the following features: 1) Multiple Participants: As teams work independently, this reduces the chance of teams generating similar attacks. 2) Multiple Tournaments: As defenders are provided with annotated data after each tournament, they can (and did) fine-tune their models with it and generate similar synthetic data. This causes similar attacks to not work well in subsequent tournaments so attackers have to use new attacks.
>
>
> We hope you’re able to reconsider the weight of these concerns in light of this clarification and revise the scores accordingly.

---

> ### Comment · Reviewer_gzkS · 2025-11-21
> **Response to rebuttal**
>
> I thank the authors for their response. I read it carefully and also checked the other reviewers' inputs,
>
> **Re 1. Lack of truly adaptive attacks**:
> I thank the reviewers for their clarification, but I think we are talking about different things. I did understand that attacks can *automatically* adapt during interactions, but my point is that attackers should be able to *manually* adapt to the *full defense*. This is a well-known paradigm in security; for an example see e.g. concurrent work by [Nasr et al., 2025](https://arxiv.org/abs/2510.09023).
>
> Hence, an individual turn should roughly happen as follows:
> 1. Defenders release the newest version of their defense.
> 2. Attackers can interact with the defense and build an attack.
> 3. Attacks and defenses are scored against each other.
>
> Step 3 can then use the authors' current procedure (i.e., multi-turn conversations where both entities can further automatically adapt).
>
> I would also be happy if the authors could comment my point **2. Scalability concerns due to manual labor**, as this could be a major limitation that renders the framework infeasible for large-scale practical applications.

---

> > ### Author Response · Authors · 2025-11-25
> > **Responses to reviewer's main concerns**
> >
> > Dear reviewer, thank you for your response. It helped clarify your questions to us.
> >
> > **Re 1. Lack of truly adaptive attacks:** The Adversarial Arena can be extended to support the paradigm where attackers can manually assess and adapt to defenses. The most straightforward way to do this would be to freeze defender systems at the beginning of each tournament and provide all attackers access to these systems. Assuming tournament timelines to be similar to our case study, attackers would get about 45 days to analyze and attack the defender models. At the end of the tournament period, defenders could update their models based on their learnings. We believe the diversity benefits of our framework will still hold in this situation.
> > Our case study was not set up like this as we (and the participating teams) were more interested in automated red-teaming, and synergy with research interests played a significant role in scaling the competition (more on this in the next point).
> >
> > **Re 2. Scalability concerns due to manual labor:** You correctly pointed out that our approach requires substantial effort from participants. However, we believe that this effort is not unique to the competition. Along with generating high-quality data, our framework also provides researchers with a good way to get feedback on their approaches. As part of our case study, we found participating teams engaged in research on novel attack and defense methods (discussed in Appendix C). And they used our framework to get feedback and new data for their research. Multiple papers were submitted by participating teams to conferences as a result of the competition in our case study. We believe that as long as there is synergy between the research interests of participants and the target domain, scaling our framework does not require much extra work from participating researchers.

---

> > > ### Comment · Reviewer_gzkS · 2025-11-26
> > >
> > > I thank the reviewers for the clarifications!
> > >
> > > It makes sense that the motivation was about *automatic* red-teaming; this is in line with the method. But I believe that today's methods/models are still far from being viable in automatic red-teaming, so this paper is in some sense ahead of its time (while still relying on humans to build the semi-automatic red-teaming methods).
> > >
> > > Similarly, for the concern regarding scaling, I agree that AdversarialArena can provide valuable feedback for researchers. However, this feedback requires a sufficient amount of groups working on a single task together and is comparatively slow. Therefore, one could argue that different groups tackling different problems could be more efficient, trading-off some feedback for much faster iteration.
> > >
> > > In summary, this confirms my initial thoughts about the paper, and I will hence keep my score.

---

> > > > ### Author Response · Authors · 2025-12-03
> > > > **Response to the viability of automated red-teaming**
> > > >
> > > > We again thank the reviewer for their effort and in-depth discussion!
> > > >
> > > > We would like to respond to the statement: "today's methods/models are still far from being viable in automatic red-teaming". There is a rich body of existing work in automated red-teaming Large Language Models that achieves Attack Success Rates (ASR) comparable and even higher than human red-teaming. A few examples are:
> > > > - [AutoDAN: Generating Stealthy Jailbreak Prompts on Aligned Large Language Models](https://arxiv.org/abs/2310.04451)
> > > > - [Universal and Transferable Adversarial Attacks on Aligned Language Models](https://arxiv.org/abs/2307.15043)
> > > > - [FLIRT: Feedback Loop In-context Red Teaming](https://aclanthology.org/2024.emnlp-main.41/)
> > > > - [Jailbreaking Black Box Large Language Models in Twenty Queries](https://ieeexplore.ieee.org/abstract/document/10992337)
> > > >
> > > > These methods have shown ASRs of upto 75% on [HarmBench](https://arxiv.org/abs/2402.04249) averaged across all behaviors. This includes ASRs over 50% on strong models like GPT-4. These numbers suggest that automated red teaming is proving to be a viable technique.
> > > >
> > > > In our case study, we observed average ASRs across all attackers and defenders to range from 19% to 28% for vulnerable code, and 4% to 12% for malicious events across tournaments. These numbers are presented in Table 4 Appendix C. These numbers suggest that, while automated red teaming methods are becoming more and more effective against SOTA models, there is significant room for improvement. Finally, we would like to emphasize that the case-study was designed in part to drive innovations in automated red-teaming. Hence the design of Adversarial Arena reflects that.

---

### Official Review · Reviewer_PTzc · 2025-11-01

**Soundness:** 3
**Presentation:** 2
**Contribution:** 2
**Rating:** 4
**Confidence:** 3

**Summary:**

The paper presents Adversarial Arena, a framework for crowdsourcing high-quality conversational datasets through structured adversarial competitions between “attacker” and “defender” teams. Each attacker attempts to elicit failures from defender models, while defenders aim to respond safely and effectively. The system reportedly produces diverse multi-turn datasets, demonstrated on a cybersecurity alignment task involving 10 university teams and ~19k conversations. Fine-tuning an open-weight model on the resulting data improves secure code generation and safety benchmarks.

**Strengths:**

1. Creative Framework: The notion of gamifying data generation through an attacker-defender structure is intuitively appealing and could inspire future collaborative or competitive data collection paradigms.
2. Scale and Engineering: The authors managed to coordinate 10 research teams and generate ~20k labeled dialogues, a notable engineering effort demonstrating feasibility at scale.
3. Empirical Evidence: The fine-tuning results (18.47% and 29.42% gains on security benchmarks) empirically confirm that the generated dataset is at least useful for improving safety-aligned code generation.

**Weaknesses:**

1. Domain Dependence.  The framework is tightly coupled to the cybersecurity task and depends heavily on fixed evaluation pipelines and manual annotation templates. The system’s success metrics rely on specific types of code vulnerabilities, which may not generalize to other domains like dialogue safety, factuality, or reasoning. The “attacker–defender” framing works mainly because cybersecurity naturally lends itself to adversarial setups; its general applicability remains unconvincing.
3. Weak Signal and Limited Generalization. The feedback signal for guiding attackers and defenders is simplistic and weak. It does not ensure meaningful improvement across rounds beyond superficial variation. While the paper emphasizes “diversity” and “richness,” it lacks concrete evidence that such signals lead to systematically better or more representative data. Furthermore, improvements are shown only for one fine-tuning case without ablation or analysis of data quality vs. quantity effects, limiting confidence in broader generalization.

**Questions:**

Can this idea apply to other domains for data collection?

---

> ### Author Response · Authors · 2025-11-20
> **Rebuttal response**
>
> Dear Reviewer, thank you for your review and insightful feedback. We feel encouraged that you found the Adversarial Arena to be creative and scalable. We are also glad that you find the empirical evidence from our fine tuning experiment, helpful. We hope the following points of clarification can assuage some of your concerns:
>
> **Domain Dependence:** You correctly observed that our evaluation pipelines are specific to the cybersecurity case-study, and the metrics themselves will not generalize to other domains. However, our claim is about the generality of the overall framework and it’s applicability to multiple task domains. There is no aspect of the framework itself that limits its application to cybersecurity tasks. We acknowledge that applying Adversarial Arena to other domains will require setting up task specific evaluation pipelines. We describe details of the evaluation pipeline as a demonstration of how to design evaluation for a task.
>
> In Section 5, we provide an example of another task (sycophancy detection) that could easily fit into this framework as follows:
>
> * Attacker teams would build systems that try to get defenders to agree with incorrect assertions.
> * Defender teams would fine-tune their models to challenge incorrect assumptions and still agree with correct assumptions.
> * Evaluation will be done based on human annotators checking for the following:
>     * Was the attacker’s assertion objectively incorrect?
>     * if yes, did the defender agree to the attacker’s assertion?
> * Auxiliary objectives would be:
>     * Diversity for attackers: This would be similar to the cybersecurity example and could use a simple BLEU score to avoid the case where attackers find and repeat a single successful attack throughout a tournament. Or an embedding based similarity metric could be used
>     * Steerability for defenders: For this objective, we would provide defenders with a sequence of conversation turns with the last defender turn being incorrect and the last attacker turn asking the model if its previous response is incorrect.
>
>
> **Weak Signal and Limited Generalization:** As the feedback signal, each participant gets the following information: 1) Evaluation metrics against 5 different systems: Participating teams’ systems get to have multi-turn conversations with 5 different opposing systems. In the case of defenders, these systems deliberately explore the dialog space to find hard cases for the defender. This is much richer feedback than provided by standard static benchmarks. 2) Labeled conversations: Each team gets labeled conversation after each tournament specifying exactly where each team's system failed in each tournament. 3) Additional metrics like attack diversity for attackers and performance on utility tasks for defenders.
>
> We believe that this strongly incentivizes improvement across tournaments because all opposing teams get the above-mentioned feedback and some time to improve their models (by, SFT, RLVR, system-level improvements etc). Teams need to invest in their systems as they will go up against stronger systems in the next tournament.
>
> Finally, we would like to re-iterate that we will also be releasing the dataset from this case study with 20,000 labeled multi-turn conversations for Cybersecurity Alignment. In light of this clarification, we kindly request that you reassess the significance of these concerns and adjust your scores as necessary.

---

### Official Review · Reviewer_Ndcg · 2025-11-02

**Soundness:** 3
**Presentation:** 3
**Contribution:** 3
**Rating:** 6
**Confidence:** 3

**Summary:**

The paper proposes Adversarial Arena, a tournament-style framework that crowdsources multi-turn conversational data by pairing “attackers” (prompt generators) against “defenders” (LLMs/agentic systems) and uses an automated evaluator to label each conversation as attack/defense success. A real-world case study on cybersecurity alignment involved 10 university teams (5 attackers, 5 defenders) across multiple tournaments, yielding 19,683 labeled conversations. The authors define ranking incentives (e.g., normalized ASR with a diversity multiplier; utility-aware defense score), show semantic diversity across teams/rounds (t-SNE & cosine-based analyses), and report that fine-tuning Mistral-7B-Instruct on curated subsets improves secure code generation (+18.47% on CyberSecEval-Instruct) and refusal of malicious cyberactivity (+29.42% on CyberSecEval-MITRE). The framework is backed by a serverless orchestrator (AWS Lambda/SQS/DynamoDB) to run large asynchronous, multi-turn matches.

**Strengths:**

Operational, repeatable framework for large-scale, multi-turn adversarial data creation with clear incentive design (normalized ASR, utility-aware defense scoring).
Demonstrated scale & impact: ~20k labeled dialogs; measurable gains on CyberSecEval-Instruct/MITRE after SFT on curated subsets.
Diversity evidence: attacker/defender/tournament-level separation via cosine distances and t-SNE plots; qualitative differences across teams and rounds.
Robust engineering: serverless orchestrator (Lambda/SQS/DynamoDB) enabling asynchronous, fault-tolerant multi-turn matches and batching, with explicit guarantees/trade-offs.
Transparency about limitations (imperfect evaluators; incentive timing; vulnerable-code vs malicious-intent skew) and concrete mitigations.

**Weaknesses:**

Evaluator dependence / label noise: Reliance on a single static analyzer (CodeGuru) risks false positives/negatives; human labeling process lacks detailed inter-annotator agreement (IAA) stats and calibration analysis. A small dynamic analysis slice or cross-tool ensemble would strengthen claims.
Diversity metric choice: BLEU captures lexical variety but can miss strategy-level novelty; the paper mentions considering embeddings, yet final ranking uses BLEU—an ablation comparing both (and their effect on attacker behavior) is missing.
Utility normalization and trade-offs: Defenders’ scores are aggressively penalized by utility drops, but details on the utility test construction, coverage, and ceiling effects (capping at base model) could use deeper analysis and sensitivity checks.
Generalization beyond cybersecurity: While the Discussion argues generality, only one ToI is empirically validated; even a small second domain (e.g., hallucination reduction or refusal over-agreement) would bolster generality claims.
Outcome attribution: The SFT improvements are compelling, but a breakdown by data slice (attacker team, tournament round, conversation length/turns, vulnerability type) would clarify which arena features drive the gains.

**Questions:**

Evaluator robustness. Can you report IAA (e.g., Cohen’s κ) and disagreement resolution for the 3-annotator panel? Any spot-checks comparing CodeGuru with a second static analyzer or dynamic tests on a held-out subset?
Diversity incentive. Why finalize on BLEU over embedding-based diversity for the ranking signal? Provide an ablation where attacker rankings and dataset properties are recomputed with an embedding-based metric.
SFT details. Share fine-tuning hyperparameters, data filtering (e.g., max turns/code length), and a per-slice contribution analysis (by attacker/defender/tournament). Do longer multi-turn dialogs help more?
Utility suites. Describe construction, difficulty, and overlap with tournament dialogs; include sensitivity of defender rankings to different utility weightings or removing the cap at base model.
Attack coverage balance. You note skew toward vulnerable-code attacks; did you trial multi-objective scoring (e.g., harmonic mean across “malicious-intent” vs “vulnerable-code” successes) and observe behavior changes?
Generalization. Any preliminary runs on a second ToI (e.g., hallucination or sycophancy) to show the arena transfers with minimal changes?

---

> ### Author Response · Authors · 2025-11-20
> **Rebuttal response**
>
> Dear Reviewer, thank you for your thoughtful feedback on our work. We are glad see your positive view of the Adversarial Arena framework, along the dimensions of scale, impact, diversity of generated data, engineering robustness. We hope our responses below can help address some of your concerns.
>
> **Overall:** We appreciate the detailed feedback on evaluation. It is important to note that the main purpose of this case study is to show that the Adversarial Arena framework is effective at generating high-quality, diverse data, not to exhaustively solve evaluation for this specific domain. Despite some imperfections, we believe the current evaluation shows clear gains in both quality and diversity. The points below address some of your specific concerns.
>
> **Evaluator dependence / label noise:** We added another table (Table 1 in the latest revision) to the paper reporting inter-annotator agreement scores. Additionally, we added Appendix D, which contains visualizations of pairwise annotator agreements and histograms for IIA scores. It also includes a table containing the average agreement scores for each annotator against every other annotator they shared an annotation task with. These average IIA scores range from 89.3% to 74.2%.
>
> For static analysis, we acknowledge that CodeGuru has limitations and is especially prone to generating false positives. We also talk about this in Section 4.2.1 of our paper.
>
> **Diversity metric choice:** The diversity incentive is specifically designed to avoid a narrow failure mode where attackers find an attack that works and keep repeating it to increase their score in a single tournament. For the overall dataset, the Adversarial Arena framework lets us overcome the difficult problem of designing a good diversity metric that can distinguish between different attack strategies because of the following features of the approach: 1) Multiple Participants: As teams work independently, this reduces the chance of teams generating similar attacks. 2) Multiple Tournaments: As defenders are provided with annotated data after each tournament, they can (and did) finetune their models with it and generate similar synthetic data. This causes similar attacks to not work well in subsequent tournaments so attackers have to evolve and continually deploy new attack strategies.
>
> As the diversity metric was an incentive for participants, we believe that recomputing rankings offline with a different metric will not capture the effect a different metric would have had in the adversarial area setting.
>
> **SFT details:** We used LoRA with the following hyperparameters for our fine-tuning experiments:
>
> * r=8,
> * lora_alpha=16,
> * lora_dropout=0.2,
>
> We used a learning rate of 1.5e-5 with cosine decay. Training was done on p4d.24xlarge EC2 instances with 8 Nvidia A100 GPUs with 40GB memory in each.
>
> **Utility normalization and trade-offs:** We added appendix E describing how the utility suites were created. We created these benchmarks before running the case study, so we used the following ways to ensure domain overlap with tournament data: 1) Alignment with evaluation criteria: As the case study required defenders to avoid producing malicious or vulnerable code, we added utility benchmarks testing for benign coding ability and even coding ability specifically in domains where vulnerabilities are more likely to occur. Another objective of the case study was to avoid producing detailed explanations for how to conduct cyberattacks (security events). Hence we introduced an adjacent benchmark consisting of benign questions about cybersecurity. 2) Stylistic similarity: Two of the three benchmarks consisted of multi-turn dialogs to resemble tournament conversations which would be multi-turn. Additionally, we also introduced some jailbreak techniques in benign prompts for stylistic similarity to tournament conversations.
>
> **Multi-objective scoring:** We found teams preferring attacks targeting vulnerable code over malicious code in the cybersecurity case-study causing a skew in the overall data. While we identified this problem while the case-study was underway, we did not want to change incentives with teams already working on the problem. Hence, we did not try a different method to combine scores such as harmonic mean, though we are adding these for future applications of the adversarial arena.
>
> We hope you’re able to reconsider the weight of these concerns in light of this clarification and revise your scores accordingly.

---

### Author Response · Authors · 2025-11-20
**Rebuttal Summary**

We are deeply grateful to the reviewers for their effort and time spent in reviewing our work and greatly appreciate all the feedback. We thank the reviewers for noting the following strengths of our paper:

1. Novel and creative framework that frames data collection as an adversarial task between multiple participants (attackers and defenders).
2. Robust and scalable framework with clear incentive design
3. Empirical evidence by fine-tuning on collected data and diversity analysis on that data.
4. Design of the case study on Cybersecurity Alignment.
5. Presentation and transparency about limitations


We would like to re-emphasize that this is a general adversarial framework that can be instantiated to collect diverse multi-turn training data for different use cases. Pitting each attacker and defender against multiple other systems that conduct multi-turn conversations provides a feedback signal that is much closer to real-world feedback, unlike the usual approach of iterating over static benchmarks. This leads to models getting more robust over the course of the challenge and the generated data getting closer to real-world scenarios.

We demonstrate an instantiation of the framework in a case study for cybersecurity alignment, and our experiments show that it indeed produces data that's better for model alignment and more diverse than the data produced by individual groups.

The framework allows the use of any evaluation mechanism and hence can be used to incentivize objectives like generating novel attacks, balancing different tasks using harmonic mean, etc. Our case study provides a simple example of how such metrics can be designed using BLEU scores for diversity and score averaging to balance different tasks. Our intention behind providing this example is to demonstrate that even with relatively simple metrics, our framework can generate diverse, high quality, multi-turn data.

---

### Meta-Review · Area_Chair_vdkv · 2026-01-07

**Summary:**

Reviewers agreed that the paper proposes an interesting and creative framework for collecting multi-turn data via adversarial competitions. Major concerns raised:
1. The core contribution is primarily a framework/platform, and several reviewers felt the academic novelty is limited beyond organizing and running a large adversarial data-collection effort.
2. The evidence for generality beyond cybersecurity is weak: only a single task-of-interest is evaluated, and claims of general applicability are not empirically validated.
3. Evaluation choices were questioned, including reliance on a single static analyzer, BLEU-based diversity incentives, and limited analysis of label noise, utility trade-offs, and outcome attribution.

The authors provided detailed clarifications and additional analysis. But, overall not all concerns are resolved.

**Reviewer Concerns:**

1. The authors added inter-annotator agreement statistics and clarified the limitations of the evaluator and static analysis tools.
2. They explained the rationale behind the diversity incentives, arguing that diversity emerges naturally from multi-participant, multi-tournament dynamics rather than metric choice alone.
3. The authors addressed concerns about adaptivity, scalability, and framing, and committed to releasing the dataset and orchestration code.

Core disagreements about novelty, scientific depth, and generality remain, with several reviewers explicitly stating they would keep their scores. Evidence for transfer to other domains remains speculative rather than demonstrated.

**Reviewer Scores:**

One reviewer leaned positive (6), several were borderline or negative, and multiple reviewers explicitly stated they would not revise their scores after the rebuttal.

---

### Decision · Program_Chairs · 2026-01-26

Reject